# Retinal Disease Diagnosis Using Deep Learning on Ultra-Wide-Field Fundus Images

**DOI:** 10.3390/diagnostics14010105

**Published:** 2024-01-03

**Authors:** Toan Duc Nguyen, Duc-Tai Le, Junghyun Bum, Seongho Kim, Su Jeong Song, Hyunseung Choo

**Affiliations:** 1Department of AI Systems Engineering, Sungkyunkwan University, Suwon 16419, Republic of Korea; 2College of Computing and Informatics, Sungkyunkwan University, Suwon 16419, Republic of Korea; 3Sungkyun AI Research Institute, Sungkyunkwan University, Suwon 16419, Republic of Korea; 4Department of Ophthalmology, Kangbuk Samsung Hospital, School of Medicine, Sungkyunkwan University, Suwon 16419, Republic of Korea; n09072@gmail.com; 5Biomedical Institute for Convergence, Sungkyunkwan University, Suwon 16419, Republic of Korea; 6Department of Electrical and Computer Engineering, Sungkyunkwan University, Suwon 16419, Republic of Korea

**Keywords:** medical image processing, deep learning, fundus image, convolutional neural network, vision transformer

## Abstract

Ultra-wide-field fundus imaging (UFI) provides comprehensive visualization of crucial eye components, including the optic disk, fovea, and macula. This in-depth view facilitates doctors in accurately diagnosing diseases and recommending suitable treatments. This study investigated the application of various deep learning models for detecting eye diseases using UFI. We developed an automated system that processes and enhances a dataset of 4697 images. Our approach involves brightness and contrast enhancement, followed by applying feature extraction, data augmentation and image classification, integrated with convolutional neural networks. These networks utilize layer-wise feature extraction and transfer learning from pre-trained models to accurately represent and analyze medical images. Among the five evaluated models, including ResNet152, Vision Transformer, InceptionResNetV2, RegNet and ConVNext, ResNet152 is the most effective, achieving a testing area under the curve (AUC) score of 96.47% (with a 95% confidence interval (CI) of 0.931–0.974). Additionally, the paper presents visualizations of the model’s predictions, including confidence scores and heatmaps that highlight the model’s focal points—particularly where lesions due to damage are evident. By streamlining the diagnosis process and providing intricate prediction details without human intervention, our system serves as a pivotal tool for ophthalmologists. This research underscores the compatibility and potential of utilizing ultra-wide-field images in conjunction with deep learning.

## 1. Introduction

Deep learning methodologies have become instrumental in reshaping diagnostic approaches in ophthalmology, heralding a paradigm where timely diagnosis and specialized treatment are prioritized. By automating complex diagnostic processes, these sophisticated models provide remarkably accurate predictions, reducing the reliance on human intervention. The fusion of deep learning techniques with ophthalmological diagnostics has garnered significant academic attention in recent years. Multiple investigations have assessed the efficacy of fundus images in detecting ocular anomalies, leveraging the capabilities of deep learning frameworks [1,2,3,4]. The synergy between these two domains underscores the transformative and far-reaching impact of such technological innovations. Beyond the scope of ophthalmology, fundus images also offer insights into systemic health conditions, including but not limited to diabetes, oncological disorders, and cerebrovascular events. This broader application is attributed to the unique anatomical characteristics of the eye, particularly the retina, which can be meticulously examined through fundus imaging, revealing critical information about blood vessels, neural structures, and connective tissues. Among the various imaging techniques, ultra-wide-field fundus imaging (UFI) is particularly noteworthy as shown in Figure 1. UFIs provide an encompassing view of the retina in a manner that is both noninvasive and user friendly. In addition, their ease of operation ensures that even individuals with basic training can proficiently acquire high-quality ocular images. Therefore, UFIs could play a significant role in telemedicine applications, proving invaluable in regions with limited access to specialized ophthalmological care, thereby facilitating remote diagnostics and prompt medical interventions.

While UFIs typically involve a higher initial investment compared to CFIs due to more sophisticated technology, they may prove more cost effective in the long term, especially in telemedicine applications. The comprehensive view offered by UFI can lead to earlier and more accurate diagnoses of peripheral retinal diseases, which can be missed by CFI. This improved diagnostic capability can potentially reduce the frequency of follow-up visits and additional testing required, resulting in lower overall costs for healthcare systems. Additionally, the ability to capture extensive retinal images in a single, noninvasive photograph makes UFI particularly suitable for remote areas, where such capabilities can drastically reduce the need for patients to travel to specialized centers. By facilitating early and accurate diagnoses, UFI can contribute to better patient outcomes and lower healthcare expenditures associated with advanced stages of ocular diseases.

Conventional fundus cameras, designed to capture the ocular surface, are limited in their scope, often capturing only between 30 to 60 degrees of the posterior pole. However, recent technological advancements have led to the development of the UFI technique, in which images can capture up to 200 degrees of the pole. A prime example of this advancement is the Optos camera, an innovative creation from Dunfermline, the United Kingdom. This camera utilizes an ellipsoidal mirror, allowing it to image the retina in various modalities, encompassing pseudocolor images, fundus autofluorescence (FAF), fluorescein angiography (FA), optical coherence tomography (OCT), and, notably, UFI. The adoption of UFI overcomes the constraints associated with conventional cameras. The advantage of UFI are twofold: (i) it extracts abundant details from scans, offering a holistic view of the retina; and (ii) the imaging procedure is notably more streamlined, for instance, it does not necessitate eye dilation, which is a standard procedure in conventional methods. The expansive practical applications of UFI are particularly noteworthy in telemedicine, offering significant benefits in clinical settings as well as remote areas with limited access to specialized ophthalmological care. In particular, UFI is crucial for diagnosing diseases such as diabetic retinopathy and retinal vascular occlusion.

The use of UFI combined with deep learning has increasingly become a focal point in ophthalmic diagnostics, with many studies validating its efficiency. In 2017, researchers began demonstrating the accuracy of machine learning models using UFI fundus ophthalmoscopy to detect rhegmatogenous retinal detachment [5], highlighting the potential for automated, high-speed diagnostics. Following this, the capability of deep learning to discern idiopathic macular holes using similar UFI imaging was established [6]. Continuing this trend, a study introduced a deep neural network method adept at identifying central retinal vein occlusion [7], leveraging the wide field of view provided by UFI ophthalmoscopy. Additional research showed the effectiveness of UFI-assisted deep learning models in detecting age-related macular degeneration [8] and treatment-naive proliferative diabetic retinopathy, thereby broadening the study of detectable conditions [9]. Moreover, the precision of deep convolutional neural networks in pinpointing retinitis pigmentosa on UFI images was explored [10], alongside the identification of lattice degeneration and retinal breaks [11], revealing the multifaceted applications of UFI imaging in retinal disease screening. Advancements in deep learning allowed for not only the detection of retinal detachment but also the assessment of macular status via fundus images. The momentum continued into 2021, with the early detection of diabetic retinopathy based on deep learning algorithms and UFI images, indicating a significant stride toward proactive retinal healthcare [12].

Recent advancements in UFI involve the identification of diseases from scans by using integrated detection, segmentation [13], and classification techniques. These studies emphasize the identification of crucial structures within the eye, such as the optic disk, macula, and adjacent regions depicted in the UFI. The recognition of these pivotal components enhances the performance of classifiers in differentiating various ocular diseases. In terms of diagnosis, extensive research has been conducted using a public eye fundus dataset to pretrain the U-net model [14], with ResNet-18 serving as its encoder. This model was later adapted to an in-house UFI dataset, in which optic disk regions are identified using the ellipse fitting method. In the classification domain, the ResNet-34 model, pretrained on the ImageNet, was employed for both training and fine tuning. This paper reviews various image processing techniques that consider the unique characteristics of UFIs, thus setting a foundation for future experimental pursuits. In 2016, the introduction of automated retinal disease assessment (ARDA) marked a significant advancement. This state-of-the-art system, powered by artificial intelligence, specializes in detecting ocular diseases, particularly diabetic retinopathy and age-related macular degeneration. Trained on a vast dataset of eye scans rigorously analyzed by experienced ophthalmologists, ARDA achieves an impressive specificity of 98.5% [15]. The combination of deep learning methodologies and medical imaging necessitates the conceptualization of an automated diagnostic assistant for UFI [16,17]. Such tools have the potential to transform ophthalmology by hastening diagnoses, minimizing human errors, and efficiently handling an increasing patient influx.

In this study, we address the challenge of predicting diseases from UFI images by utilizing state-of-the-art deep learning methodologies. Our approach first uses a specialized preprocessing procedure to augment the decision-making prowess of the AI system. This approach employs cutting-edge models pretrained on ImageNet [18], acclaimed for their performance in image classification tasks. This selection encompasses formidable models such as ResNet152, Vision Transformer, InceptionResNetV2, RegNet, and ConVNext, all of which we integrated with our dataset for disease diagnosis. The results show that the proposed model achieved an impressive AUC score, with a peak of 96.47%. However, our investigation extended beyond model deployment; we performed a comparative analysis on the performance of all aforementioned models to identify the optimal model to perform the required task. By harnessing the capabilities of the top-performing model, we crafted a visualization technique that elucidates prediction determinants. This visualization approach, highlighting regions within images suggestive of the diseases, offers critical insights into disease localization within the eye, and it is crucial for both the ongoing research and clinical evaluations. The key contributions and findings of our study are summarized as follows:Training and evaluation of deep learning models: We employ a range of state-of-the-art deep learning models, such as ResNet and Vision Transformer, to classify images and predict the presence of eye diseases. These models are compared to discern their performance metrics, enabling us to identify the model that yielded the most promising results in disease prediction.Output visualization for in-depth analysis: To ensure that our findings are transparent and interpretable, we provide visual representations of our results. This includes heat maps which precisely indicate the focal regions in images displaying signs of diseases where evident. These visual aids are invaluable for researchers and medical professionals seeking to understand the exact locations and extents of potential disease manifestations.

## 2. Materials and Methods

### 2.1. Data

In this study, we utilized an in-house dataset comprising 4697 images. These images were retrospectively amassed from 2 January 2006 to 31 December 2019. They represent a diverse group of patients—both with and without retinal diseases—who underwent eye examinations at the Kangbuk Samsung Hospital Ophthalmology Department during this period. A critical step in our data-handling procedure was ensuring the privacy and confidentiality of the patients involved. To this end, upon acquisition of the images, all identifiable information, including patient IDs, was meticulously removed. Subsequently, these anonymized images were dispatched to a medical professional for accurate labeling. Our study rigorously adhered to the tenets outlined in the Declaration of Helsinki. Furthermore, our protocol underwent scrutiny and received approval from the Institutional Review Boards (IRB) of Kangbuk Samsung Hospital under the identifier No. KBSMC 2020-01-031-001. As our study is retrospective and delves into medical records with all data being fully anonymized, the IRB graciously waived the requirement for informed consent from the patients.

Our dataset, comprising images with a resolution of 2600 × 2048 pixels, was categorized into normal and abnormal images. The “normal” category encompasses images depicting eyes free from any discernible disease, whereas the “abnormal” category is a collection of images displaying eyes afflicted with one or more conditions. These images were also classified by our professional ophthalmologist. Among the normal images, 1444 and 161 healthy eye images were utilized for categorizing into training and testing sets, respectively. In contrast, among the abnormal images, 2782 and 310 were allocated as training and testing images, respectively.

### 2.2. Proposed System

Figure 2 presents the proposed framework, comprising three essential phases: data augmentation, image-quality enhancement, and disease classification. A prevalent challenge in medical imaging research is the limited availability of labeled images. Such a limitation could mainly be attributed to the complexities associated with acquiring patient consents, thereby elevating the significance of each labeled dataset. To address this, we initially performed data augmentation to enrich the dataset. This process not only increases the dataset volume but also instills diversity, facilitating better model generalization and mitigating overfitting risks. Next, each image was thoroughly enhanced for quality, which is crucial for refining the image attributes, thereby ensuring that the neural network is presented with salient features devoid of extraneous noise or distortions. After being processed, these images were converted into tensors, making them ready for deep neural network processing. Finally, state-of-the-art neural networks were deployed for disease classification. A distinguishing characteristic of our framework is its fully automated nature, eliminating the need for human interventions, thereby reinforcing our commitment to establishing an autonomous and accurate system for ocular disease diagnosis.

The dataset, sourced from our hospital, comprises high-resolution images with dimensions of 2600 × 2048 pixels. Although such high-definition images offer intricate details beneficial for certain analyses, their direct incorporation into deep learning frameworks presents computational challenges. A primary concern is the significant computational overhead inherent in processing these resolutions. The direct use of such images for model training is not only computationally burdensome but also risks extended training durations and potential memory constraints. Therefore, the differential image interpretation between deep learning algorithms and human clinicians must be determined. Whereas medical professionals might focus on subtle diagnostic patterns within a high-definition image, neural networks process extensive numerical pixel arrays, extracting patterns from these vast data. This voluminous detail can inadvertently obscure essential patterns, potentially diminishing the model’s efficacy. To synchronize the need for detail with computational pragmatism, the images were resized to a resolution of 512 × 512 pixels by using bi-linear interpolation—a method that employs successive linear interpolations in both the x and y directions.

#### 2.2.1. Data Augmentation

State-of-the-art deep learning models often feature an intricate architecture, boasting multiple convolution layers. These deep architectures, with numerous parameters, enable models to grasp more intricate and distinct representations of data. However, such complexities have certain limitations, i.e., these models demand vast amounts of labeled data for effective training and validation. Such vast amounts of data are unavailable in the field of medical imaging, primarily because of the stringent confidentiality obligations surrounding patient data. To address this data scarcity, we leveraged data augmentation—a suite of computer vision techniques to artificially expand our dataset. The principle of these techniques is to produce variants of the original images, while preserving their inherent medical information and characteristics.

Horizontal and vertical flipping: By transposing the image, i.e., by swapping the *x* and *y* indices, we can achieve both horizontal and vertical flipping. This straightforward operation effectively doubles the dataset, providing mirrored versions of each image.
(1)$$\begin{array}{c} \left[\begin{array}{c} x^{\prime }\\ y^{\prime }\\ 1 \end{array}\right]=\left[\begin{array}{ccc} -1 & 0 & w\\ 0 & 1 & 0\\ 0 & 0 & 1 \end{array}\right]\left[\begin{array}{c} x\\ y\\ 1 \end{array}\right] \end{array}$$
(2)$$\begin{array}{c} \left[\begin{array}{c} x^{\prime }\\ y^{\prime }\\ 1 \end{array}\right]=\left[\begin{array}{ccc} 1 & 0 & 0\\ 0 & -1 & h\\ 0 & 0 & 1 \end{array}\right]\left[\begin{array}{c} x\\ y\\ 1 \end{array}\right] \end{array}$$

Rotation: This is achieved through affine transformation, a type of geometric transformation that can retain straight lines and parallelism, albeit at the expense of the absolute preservation of distances and angles. The mathematical underpinning of the rotation technique can be denoted as:
(3)$$\begin{array}{c} \left[\begin{array}{c} x^{\prime }\\ y^{\prime } \end{array}\right]=\left[\begin{array}{cc} \cos\theta & -\sin\theta \\ \sin\theta & \cos\theta \end{array}\right]\left[\begin{array}{c} x\\ y \end{array}\right] \end{array}$$
where (x’,y’) denotes the new data point post-rotation, (x,y) symbolizes the original data point, and θ represents the rotation angle. This transformation is methodically applied to all coordinates within an image to produce the rotated version. A salient feature of our augmentation strategy is its randomness. This stochastic approach, whether to flip an image (with a probability of 0.5) or select a random rotation angle, serves a dual purpose: (i) it ensures a diverse dataset, making it more representative, and (ii) it actively combats the specter of overfitting, ensuring our model does not overly familiarize with the training data but remains generalizable to new, unseen data.

#### 2.2.2. Preprocessing

Before initiating the training process, our dataset was subjected to a preprocessing procedure to optimize model efficacy. To enrich our data repository, we utilized data augmentation strategies that encompassed random horizontal and vertical flips in conjunction with random rotations. This approach generates various pixel configurations while retaining the intrinsic attributes of the original images, effectively expanding our dataset to a count of over 21,000 images. By considering the inherent variability, often characteristic of medical images, we incorporated image enhancement methodologies. Utilizing the image histogram, we modulated brightness to achieve harmonized illumination and heightened the overall contrast to guarantee a homogenized pixel intensity distribution. This process accentuated the features vital for accurate medical interpretation. To ensure computational tractability and model congruence, images designated for both training and testing were resized to a consistent resolution of 512 × 512 pixels. Subsequently, normalization was applied to standardize the pixel intensities, priming them for optimal neural network training.

Although capturing UFI is noninvasive and convenient, the quality of these images is unsatisfactory (in terms of image property), and the images often include artifacts (such as camera light) or other body parts (such as eyelashes). These elements could be a problem for deep learning, as the models consider them as an aspect of the image that must be considered. To overcome these problems, some image enhancing techniques have been applied, such as histogram equalization or brightness/contrast adjustment. The histogram equalization technique was employed to equally distribute the intensity values of the images. In addition, the brightness/contrast was adjusted according to image pixel intensity *I*, and the resulting intensity can be represented as:(4)$$I^{\prime }=\alpha \times I+\beta$$
where *I* is the original pixel intensity, $I^{\prime }$ is the pixel intensity after adjustment, α is the contrast control, and β is the brightness control. Note that α>1 represents an increase in the contrast value, and 0<α<1 represents a decrease in the contrast. An α of 1 will leave the contrast unchanged. Positive values of β represent an increase in the brightness, whereas negative values represent a decrease in it.

#### 2.2.3. Deep Learning Classification

Over the past few years, the deep learning field has witnessed intensive research on image classification based on deep learning architectures [19,20,21]. Conventional architectures, including VGG, CenterNet, and ResNet, historically dominated the computer vision landscape. However, the introduction of Vision Transformer (ViT) has significantly increased discussions within the field of deep learning. The primary advantage of ViT is its attention mechanism, which adeptly prioritizes salient portions of input data, resulting in the exemplary performance of diverse tasks [22]. Earlier, convolutional neural networks (CNNs) formed the foundation of image analysis in computer vision [23]. These networks, characterized by a series of interconnected neurons, can easily extract intricate features from images, subsequently synthesizing meaningful representations. However, the introduction of the Transformer architecture heralded a noteworthy evolution in this domain. An increasing number of researchers are now investigating this architecture because of its demonstrated proficiency in various benchmark tasks, often surpassing the capabilities of conventional CNNs [24,25]. Notably, in some studies [26], ViT distinctly outperformed its CNN counterparts, particularly in tasks centered around disease detection and grading, highlighting its potential in discerning critical features within medical images. In this paper, we tested five of the most recent and state-of-the-art methods in computer vision, namely ResNet152, ViT, Inception-ResNet-v2, RegNet and ConVNext, for classifying the UFI dataset. The core blocks of these models are shown in Figure 3.

Since the introduction of ResNet [27], it has rapidly superseded the prevailing model VGG [28] not only in terms of accuracy but also computational efficiency. Conventional wisdom within the deep learning community posited that the deepening of neural networks—by incrementing layers—would inherently bolster model performance. However, this technique poses several complications. As networks deepen, they increasingly encounter the vanishing gradient effect, which complicates the training of profound networks and culminates in diminished performance outcomes. ResNet provides an innovative resolution to this predicament. Its main concept is the “residual block”, premised on the “Identity Shortcut Connection”. Mathematically, a residual block can be articulated as:(5)$$\begin{array}{c} y=F(x,\left\{W_{i}\right\})+x \end{array}$$
where *x* and *y* represent the input and output of the respective layers, while $F(x,\left\{W_{i}\right\})+x$ represents the residual mapping to be ascertained. This framework permits certain network layers to avoid one or multiple subsequent layers, forging a more succinct traversal path through the network. The underlying thesis suggests that, in specific contexts, optimizing the residual mappings is more effective than directly adjusting the original mappings. Based on this paradigm, ResNet assured that the augmenting network depth did not degrade the overall performance. These “shortcut” or “skip connections” preserve the advantages of intricate architectures, such as discerning nuanced data patterns, while obviating the typical adversities of the profound networks. Consequently, ResNet models, with their multitudinous layers, are adeptly trained without capitulating to the aforementioned challenges, thereby redefining the benchmarks in diverse computer vision tasks.

The introduction of ViT allowed the use of architectures previously reserved for natural language processing in the field of computer vision. The ViT draws its strength from the attention mechanism, which enables it to selectively focus on salient regions of the input, thereby enhancing the interpretability and relevance of its outputs. At the core of this attention mechanism are three pivotal elements introduced in [22], namely the query (Q), key (K), and value (V). When combined with positional encoding, these elements facilitate the learning of attention weights within an encoder–decoder framework, ensuring that spatial relationships within the input are retained. By adapting the original Transformer structure for visual tasks, the ViT inherits its predecessor’s properties, ensuring the continued benefits of the original design. The most notable modification in ViT is the treatment of the input. Instead of receiving raw pixel values, ViT utilizes images as sequences of patches. These patches, derived by segmenting the input image, can be mathematically represented as:(6)$$\begin{array}{c} x_{p}\in R^{H\times W\times C}\to x_{p}\in R^{N\times (P^{2}\cdot C)} \end{array}$$
where (*H*, *W*) is the resolution of the original images, (*C*) is the number of channels, (*P*, *P*) is the resolution of each image patch, and N=HW/P2 is the number of patches. After these patches are flattened into sequences, they are converted into embeddings through learnable parameters. These embeddings are then further processed, with a classification head appended to predict the output class. Notably, this head uses a multilayer perceptron (MLP) during pretraining and a linear layer in the fine-tuning phase.

Inception-ResNet-v2 is a CNN architecture that was built using the Inception structure but further fused with a residual network, which is the most important feature of a ResNet architecture. Originally stemming from the InceptionV3 framework, Inception-ResNet-v2 incorporates key features from both designs, resulting in a CNN with 164 layers. This model was pretrained on a diverse set of data spanning 1000 categories, designed to accept inputs of dimensions 299 × 299 pixels. The guiding principle of the Inception architectures is to favor width over depth. This is embodied in the ‘inception blocks’, which concurrently execute convolution operations of varied sizes within a singular layer. Based on these operations, dimensionality was reduced to ensure computational efficiency. What sets Inception-ResNet-v2 apart is the integration of Residual Inception Blocks. Within these blocks, each convolution from the inception units is harmoniously melded with a residual connection, a concept central to the ResNet design. The Inception module is defined as a combination of convolutional filters of different sizes applied to the same input layer. Assume I(x) represents an inception block operation on input *x*. Then, the output of this operation would be I(x). In the context of the residual connection, the output is obtained by adding the original input *x* to the output of the inception operation I(x).

RegNet [29] represents a new approach to dynamically adapting neural network architectures through parameterization. Instead of maintaining static architectural dimensions, RegNet leverages parameterization to describe the widths and depths of networks by using a quantized linear function. The development process is initiated by establishing a comprehensive space called “AnyNet”, encompassing a broad spectrum of unconstrained network architectures. In this study, each of these networks underwent rigorous training and evaluation to filter out the most efficient ones. This allows to reduce the complex space of possibilities so as to extract the critical parameters governing the best models. The use of such a selective process resulted in the RegNet design space, characterized by a regular and systematic network structure.

So far, we tested two types of deep learning methods for classification, namely CNNs (ResNet152, InceptionResNetV2, and RegNet) and a Transformer (ViT). However, recent studies have focused on a model that inherits the most compelling features of both methods: ConVNext. Introduced in 2022, ConVNext is a CNN that is significantly inspired by the design philosophy of ViT. Instead of being another convolutional model, the ConVNext presents itself as a “modernized” convolution network, morphing the foundational principles of a standard ResNet to align more closely with the architectural nuances of ViT. Its innovative design not only pays homage to its predecessors but reportedly surpasses the original designs in performance metrics [30]. Table 1 shows the characteristics of each model, where ConVNext is the largest model in size (337.95 MB) and has the most parameters among all the models (88,600,000 parameters).

### 2.3. Measurement Metrics

In evaluating our models, we utilized several standard measurement metrics:AUC score: The Area Under the Receiver Operating Characteristic Curve (AUC-ROC) measures a classifier’s ability to distinguish between classes.F1 score: This is the harmonic mean of precision and recall, balancing the trade-off between the two.Kappa score: Cohen’s Kappa score measures the inter-rater reliability for categorical items, indicating the precision of the classifier.

Each of these metrics provides a different perspective on the performance of our models, from their discriminative power (AUC) to their balance between precision and recall (F1), and their reliability (Kappa).

### 2.4. Implementation Details

The use of complex deep neural networks, including architectures such as ResNet or Transformer, could often result in the limited availability of adequately labeled training data. A widely endorsed mitigation, especially pertinent to medical imaging, is the use of transfer learning. This methodology facilitates a model to extrapolate insights from one domain and proficiently repurpose them for another cognate task within a similar purview. The success of transfer learning can be attributed to comprehensive labeled databases, such as ImageNet or Cifar, which serve as a foundation for the preliminary training of these extensive networks. By harnessing these corpora, models can undergo a pretraining regimen, thereby optimizing their initial weight configurations. These models are then refined using domain-centric datasets, such as UFI, ensuring task specificity while capitalizing on the generalized image semantics. Such a strategy not only bolsters the model’s efficacy and expedites convergence but also reduces reliance on voluminous labeled data, resulting in significant reductions in computational overheads and epoch durations. Table 2 lists the comprehensive training parameters. Our empirical assessments substantiate the merits of transfer learning. Specifically, a model primed with pretrained weights manifested an AUC score at the start of training (85.6%) better than that of a model without such a pre-configuration (65.4%).

Our proposed methodology utilizes Pytorch [31], operating on GeForce RTX 3080 Ti GPUs. The computational prowess of these GPUs is pivotal, handling both the extensive dataset and the intricate nature of the model. By considering the parallel processing abilities of the GPUs, we can efficiently train our model and accelerate the pretraining processes by using the recommended methods. Performance measurements are calculated automatically by using the sklearn libraries.

## 3. Results

To validate our proposed system, we assessed the performance of each model by employing three distinct methods: processing raw data, data augmentation (without pre-processing), and deploying the proposed system (with both augmentation and pre-processing). Figure 4 illustrates the ROC curves for the five models under study: ResNet152, InceptionResNetV2, RegNet, ConVNext, and ViT. Within these graphs, the yellow line represents models trained on raw data, the blue line represents models trained using only augmentation, and the red line represents models trained using our proposed system. Among the models, ResNet152 and ConVNext emerged as top performers, registering AUC values of 96.47 (with a 95% confidence interval (CI) of 0.953–0.975) and 96.13 (with a CI of 0.948–0.973), respectively. Their high performance results were highlighted by their proximity to an AUC of 1. For clarity, all evaluation metrics were determined based on the configurations detailed in Table 3. Specifically, under these settings, InceptionResNetV2, RegNet and ViT achieved average AUCs of 95.2, 96.04 (95% CI of 0.947–0.972), and 95.2 (95% CI of 0.937–0.967), respectively.

More detailed performance results of each models are shown in Table 3. We computed each models’ evaluation scores, i.e., AUC score, F1 score, Kappa score, and accuracy (Test accuracy). As shown, ResNet152 achieved the best accuracy of 89.17%, followed by ConVNext. In addition, ResNet152 achieved the highest scores among all the classifiers (4/4 metrics), with the highest F1 and Kappa score of 89.09% and 75.61%, respectively. Note that in our dataset, the fundus images contain many diseases, including age-related macular degeneration, diabetic retinopathy, epiretinal membrane or retinal vein occlusion. A single image could comprise more than one disease, complicating the appropriate prediction of abnormal images in the dataset. Nonetheless, our approach still achieved practicable results for this task.

For models trained on raw data, ConVNext demonstrated superior performance metrics with an AUC score of 94.44, F1 score of 87.04%, Kappa score of 71.10%, and accuracy of 87.05%. ResNet152 trailed closely, whereas ViT lagged in comparison, with notably lower metrics overall. In particular, it achieved an AUC score of 85.0. When integrated with data augmentation techniques, a tangible enhancement was observed in the performance metrics for most models. ConVNext again emerged at the forefront with a commendable AUC score of 96.07, F1 score of 88.71%, Kappa score of 77.45%, and accuracy of 89.06%. Notably, ViT showed improvement, especially in term of its Kappa score, displaying a significant leap to 50.03 from 42.08 when using raw data.

We examined the performance of five deep learning architectures during training by plotting the loss curves of these models as shown in Figure 5. The loss trajectories, charted over training epochs, offer a lucid depiction of the learning efficacy and dependability of each model. All architectures displayed a diminishing loss trend, suggesting resilience against overfitting and endorsing the validity of the reported outcomes. Notably, ResNet152 and ConVNext emerge as frontrunners with minimal loss magnitudes. This metric accentuates the distinguished capability of these two models to generalize and discern complex patterns within the dataset. Although the remaining architectures also exhibit commendable performance, the marked proficiency of ResNet152 and ConVNext highlights their potential applicability for applications that necessitate paramount accuracy and consistency.

To enhance the transparency and comprehensibility of our deep learning models, we illustrate the post-training heatmaps in Figure 6. Sequentially, each column displays the resized input image, succeeded by the heatmaps generated from ResNet152, ViT, RegNet, InceptionResNetV2, and ConVNext. The subsequent rows represent original images juxtaposed with their corresponding heatmaps as interpreted by each model. In particular, the top three rows represent eyes with diagnosed pathologies, whereas the final row represents a healthy eye for comparative analysis. These heatmaps employ a color spectrum from blue to red superimposed on the original images, with the red regions emphasizing the predominant focus during model predictions. Preliminary inspection suggests that the models consistently localize anomalies in the images identified as pathological. This localization is based on a comparative evaluation with respect to healthy eyes, displaying distinct irregularities in the pathological eyes. To bolster the credibility of these observations, we consulted two experienced ophthalmologists. Their insights were revelatory, pinpointing a notable alignment between the areas emphasized by ResNet152 and their diagnostic focal points, particularly in the posterior ocular segments. According to their professional knowledge, these regions frequently harbor pivotal signs indicative of the ocular health status. Although this information cannot be considered conclusive evidence of our models’ interpretive efficiency, the expert feedback underscores the promising efficacy of these AI-driven methodologies.

In the subsequent evaluation, we visualized the feature maps extracted from each model after the application of 2D convolutional layers as depicted in Figure 7. These feature maps serve as a window into the complex learning mechanisms of the networks, revealing the progression of feature extraction and abstraction at various depths. For multi-layered architectures, such as ResNet152, InceptionResNetV2, and RegNet, we encounter a vast array of convolutional layers. To provide a meaningful visualization without overwhelming detail, we judiciously selected and presented feature maps from three or four representative layers, chosen based on their relevance to capturing pivotal features in the UFI data. Conversely, for ConvNext, we offered a comprehensive visual array, detailing the feature maps following each convolutional layer to demonstrate the model’s methodical construction of features from simple to complex. Furthermore, with the Vision Transformer (ViT), we concentrated on showing its class activation mappings, which highlight the critical areas within the input image that contribute most significantly to the model’s classification decision. Across all models, these visualizations provide a deeper understanding of the discriminative regions that the models prioritize during their learning process. The detailed terminal feature maps are particularly revealing, as they allow us to pinpoint disease-associated regions within the medical images with high precision. This accuracy is notably prominent in models like ResNet, ViT, and ConvNext, showcasing their proficiency in discerning pathological features. Those insights underscore the capabilities of these models in medical diagnostic applications, where the precise identification of abnormalities is crucial.

Figure 8 illustrates the t-SNE visualizations, which provide an intuitive representation of the high-dimensional feature spaces learned by various deep learning models. These visualizations extract complex data patterns into two-dimensional plots that can be readily interpreted, allowing us to assess the models’ abilities to segregate classes spatially. In the case of ResNet152, InceptionResNetV2, and ConVNext, we observe distinct, well-separated clusters, indicative of their superior pattern extraction and classification capabilities. The clear demarcation between ‘normal’ and ‘abnormal’ classes in their respective t-SNE plots underscores the effectiveness of these architectures in capturing and emphasizing the salient features necessary for accurate class differentiation. On the other hand, some models exhibit less defined clustering, with overlaps in the t-SNE space, suggesting potential challenges in distinguishing between classes with high precision. These patterns are pivotal, as they reveal the intrinsic discriminative power of each model, with the spatial separation of the clusters serving as a proxy for the models’ ability to generalize from the data. This visual comparison of t-SNE embeddings from multiple models not only demonstrates the robustness of certain architectures in feature representation but also affirms the critical impact of feature extraction proficiency on the overall performance of the models. The discernible differences in cluster formation highlight the importance of model selection in medical diagnostic applications, where the accuracy and reliability of predictions are of the utmost importance.

In the subsequent visual representation, the model-based predictions for each input image are shown in Figure 9. For every input, the model depicts not only the prediction but also its true label, presenting a clear insight into the model’s accuracy. Similarly, the “model confidence” is a crucial factor in understanding the certainty of the model prediction. This factor can be fundamentally derived from the model’s softmax function, which is defined as:(7)$$\begin{array}{c} P\left(y_{i}\right)=\frac{e^{y_{i}}}{\sum_{j=1}^{K}e^{y_{j}}} \end{array}$$
where $y_{i}$ represents the raw score (logit) for class *i*, and *K* is the total number of classes. The output of softmax transforms these logits into probability values. For instance, if the model predicts an image as “abnormal” and assigns it a softmax probability of 0.95, the model represents a 95% confidence level. Similarly, a “normal” prediction with a softmax value of 0.85 translates to 85% confidence in the image being “normal”. Although these confidence scores can be highly suggestive of the model’s conviction, they do not always mirror its actual accuracy or reliability. Such discrepancies could be attributed to issues such as biased training data or overfitting.

Figure 10 plots the inference times with respect to the number of parameters for each model. As observed, InceptionResNetV2 demonstrated the shortest inference time and the fewest parameters, indicative of its capability to process data efficiently with a streamlined architecture. Nevertheless, this efficiency seems to compromise its overall performance as demonstrated by its relatively suboptimal results. In contrast, ResNet152 achieves a remarkable equilibrium: it demonstrates a near-minimal inference time with a moderate number of parameters. Overall, it is a top performer among the models in terms of accuracy. This underscores the efficacy of its architectural design in balancing speed and precision. ConVNext presents a distinct profile: despite comprising the most extensive parameter set, its inference time remains competitively low. Such a profile suggests adept utilization of its expansive parameters for swift predictions. This examination underscores the varying design philosophies and the consequential trade-offs among the model complexities, computational speeds, and performance, emphasizing that an increased parameter count is not a straightforward indicator of either prolonged inference or enhanced performance.

## 4. Discussion

Our findings demonstrated the potential of utilizing deep learning models in the field of UFI for diagnosing eye diseases. The high AUC score achieved using deep learning is particularly noteworthy, as it is a strong indicator of the model’s capability to distinguish between diseased and healthy conditions. In real-world clinical applications, such a high AUC indicates AI to be a reliable diagnostic tool that can minimize false positives and negatives, thereby ensuring that patients receive timely and appropriate care. This is especially crucial in settings where there is a dearth of experienced ophthalmologists, as an automated system with high accuracy can ensure quality care. The integration of AI techniques in the medical domain could pave the way for significant advancements as evidenced by the robust results yielded in our study. These methods hold great potential for augmenting clinicians’ diagnostic abilities, expediting patient assessments, and, consequently, accelerating the treatment initiation process. The evolution of such deep learning systems increases the possibility of the development of autonomous medical instruments that could perform diagnoses without direct human intervention, promising both cost- and time effectiveness in the healthcare process.

Our study introduces several advancements in the application of deep learning models to ultra-wide-field fundus imaging (UFI) datasets yet acknowledges key limitations. Firstly, the initial exploratory phase, which included models trained on unprocessed images, was not systematically documented, limiting our ability to empirically validate the impact of the preprocessing steps, such as histogram equalization, on the model accuracy. This gap in documentation and analysis hinders a full understanding of the preprocessing’s contribution to enhancing deep learning performance in UFI contexts, an aspect we aim to address in future research. Additionally, the unique approach of employing UFI for a binary ‘normal’ vs. ‘abnormal’ classification presents challenges in benchmarking against existing studies, which typically focus on distinguishing ‘normal’ from specific diseases. This novel classification approach, while innovative, affects the direct comparability and interpretability of our models, underscoring the need for future work to establish a baseline for such broad-spectrum classification using UFI. As the field evolves, we believe our study will provide a valuable reference for subsequent research in this area, contributing to the broader understanding and application of UFI in ophthalmic diagnostics.

Our research primarily focuses on the technical aspects of image analysis and model development, which means the direct correlation with clinical outcomes and patient-oriented evidence is limited. The clinical applicability of our findings requires further validation through prospective clinical trials and collaboration with medical practitioners to ensure that our models align with the clinical realities and patient needs. Secondly, while our models demonstrate promising results in image analysis, we did not explore the integration of these diagnostic capabilities with treatment strategies, particularly advanced drug delivery systems, like controlled release methods. The omission of this aspect limits the scope of our research in providing a comprehensive view of patient care that encompasses both diagnosis and treatment. Addressing these limitations in future work will be crucial for transitioning from technical innovation to practical, patient-centered applications in ophthalmology.

Nevertheless, the task of disease diagnosis using medical images is an ever-evolving field with ample scope for further exploration. Given the inherent challenges of data labeling, especially the need for expert input in the case of medical images, researchers must consider studying cutting-edge deep learning techniques that can deliver results with minimal labeled data. In such scenarios, techniques such as semi-supervised and self-supervised learning [32] are promising candidates. This is because they do not necessitate a vast corpus of labeled data to discern the intricate feature representations in medical images. Such techniques could dramatically reduce the need for expert interventions in the training process yet, as preliminary studies suggest, still maintain, or even exceed, the current performance benchmarks in medical imaging.

## 5. Conclusions

This study proposed an approach combining state-of-the-art deep learning models with UFIs. Compared to conventional fundus images, the proposed method achieved competitive performance in the diagnosis of disease based on retinal images. Along with the development of CNNs and the attention module, these deep learning algorithms could extract features and demonstrate important regions of the medical images showing signs of lesions and hemorrhage. This study exploited the performance of supervised learning models such as ResNet or Transformer on the UFI dataset, accurately understanding how well an AI system would be able to achieve generalization with this modal of medical images. In the future, we plan to develop new self-supervised learning methods to tackle the deficiencies in labeled medical images.

## Figures and Tables

**Figure 1 diagnostics-14-00105-f001:**
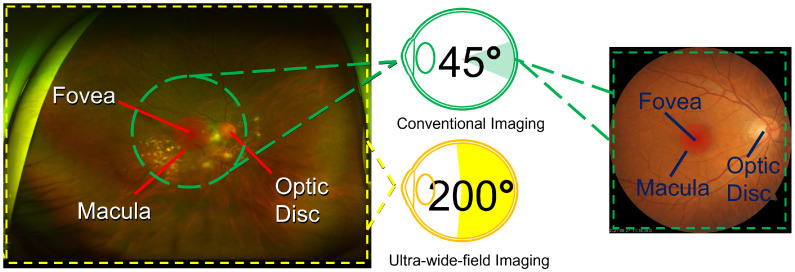
Comparative view of ultra-wide-field fundus imaging (UFI, left) versus conventional fundus imaging (CFI, right), showcasing the extent of retinal coverage and detail resolution. UFI captures a comprehensive field of view, revealing the fovea, macula, and optic disk alongside peripheral retinal details that are not visible in the narrower field of CFI, which focuses on central retinal structures with greater detail. This contrast highlights the diagnostic advantages of UFI in assessing peripheral retinal pathology and the detailed visualization of central retinal features by CFI.

**Figure 2 diagnostics-14-00105-f002:**
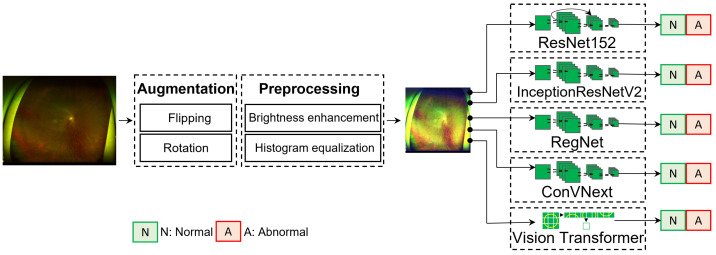
Deep learning-aided eye disease diagnosis system. The images first undergo augmentation and preprocessing. Next, the preprocessed images are fed into deep learning neural networks to learn feature representations. The model then outputs the predictions as normal or abnormal eyes.

**Figure 3 diagnostics-14-00105-f003:**
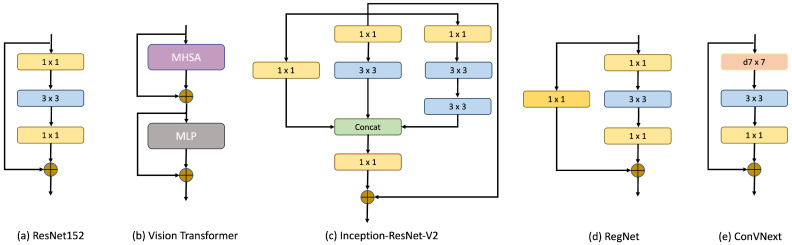
Different core blocks of the proposed deep learning models. ResNet152 employs convolution layers of size 1 and 3, RegNet introduces a design space with additional convolution layers of 1×1, and ConVNext introduces a depth-wise convolution. Vision Transformer comprises Multi-head Self-attention (MHSA) and multilayer perceptron (MLP).

**Figure 4 diagnostics-14-00105-f004:**
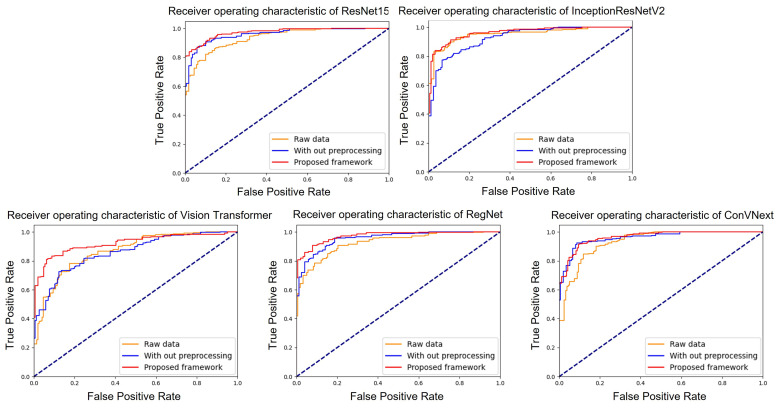
AUC score of ResNet152, Vision Transformer, InceptionResNetV2, RegNet and ConVNext.

**Figure 5 diagnostics-14-00105-f005:**
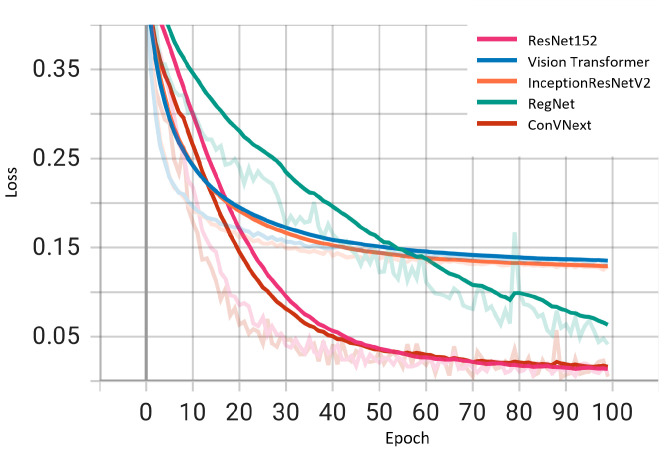
Training losses of ResNet152, Vision Transformer, InceptionResNetV2, RegNet and ConVNext. The transparent lines represent the raw training loss for each epoch, while the solid lines show the smoothed training loss with a smoothing factor of 0.6.

**Figure 6 diagnostics-14-00105-f006:**
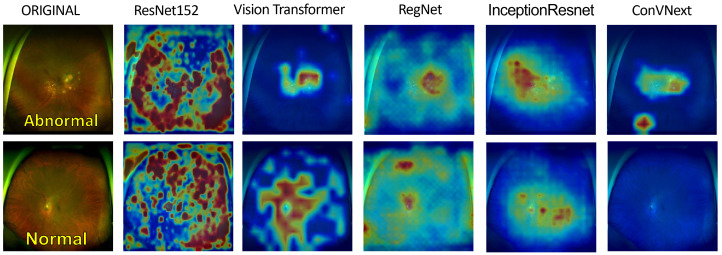
Heatmaps generated from each deep learning model. The first-row images display abnormal eyes, and the second-row images represent normal eyes.

**Figure 7 diagnostics-14-00105-f007:**
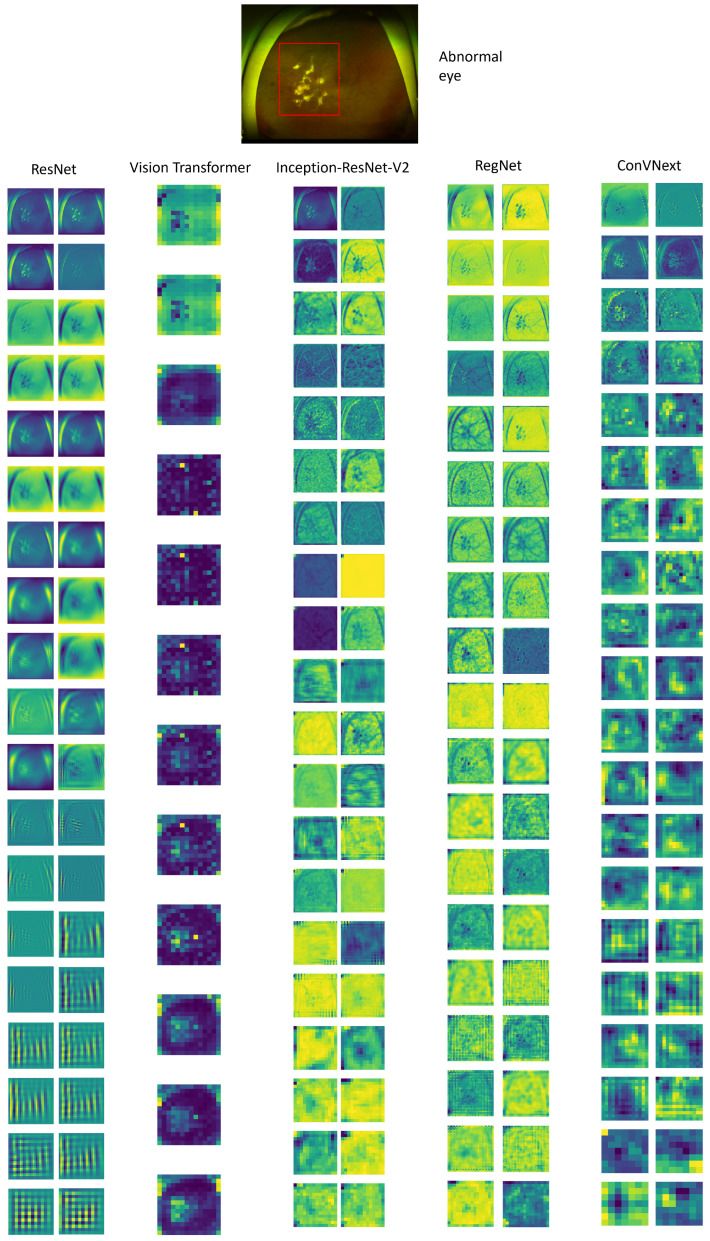
Feature map generated from each deep learning model after each convolution layer (or after each block of attention, in the case of Vision Transformer).

**Figure 8 diagnostics-14-00105-f008:**
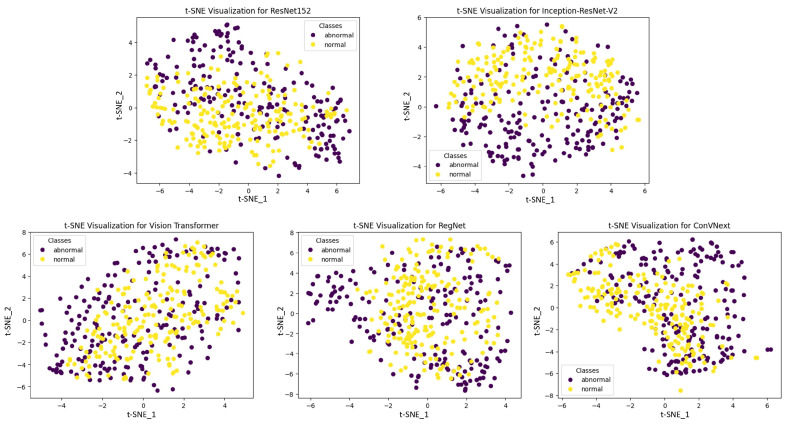
t-SNE visualization for each deep learning model, generated from 400 samples (200 for each class).

**Figure 9 diagnostics-14-00105-f009:**
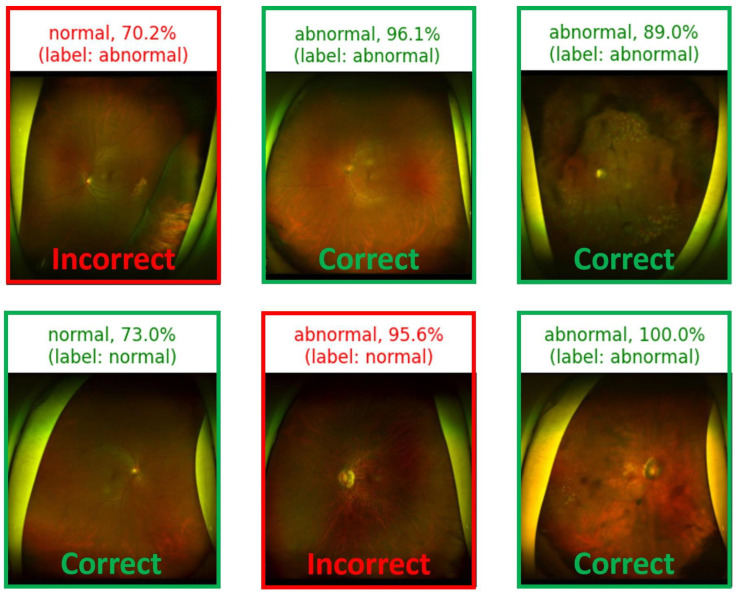
Examples of ResNet152 model predictions, given an input image.

**Figure 10 diagnostics-14-00105-f010:**
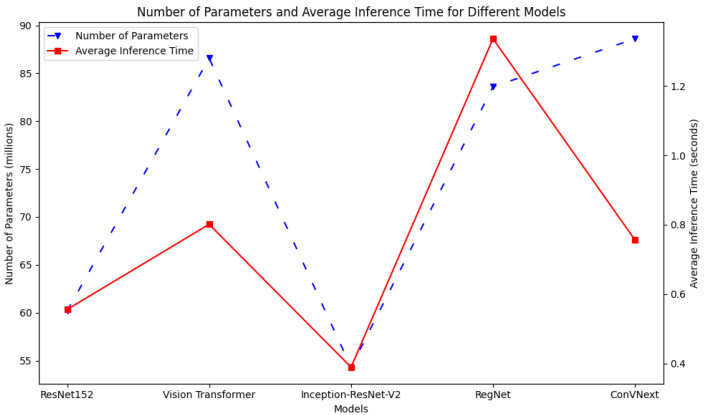
Inference time for each model (average time taken for each model to generate a prediction, given an input image).

**Table 1 diagnostics-14-00105-t001:** Characteristics of deep learning models.

Models	Size (MB)	Number of Layers	Total Number of Parameters
ResNet152	230.19	152	60,192,808
Vision Transformer	330.23	-	86,567,656
InceptionResNetV2	207.41	572	54,309,538
RegNet	319.34	293	83,590,140
ConVNext	337.95	300	88,600,000

**Table 2 diagnostics-14-00105-t002:** Training configurations of deep learning models.

Model	ResNet15, InceptionResNetV2, RegNet, ConVNext	Vision Transformer
Image size	512 × 16,512	224 × 16,224
Batch size	64	32
Number of epochs	100	100
Learning rate	0.001	0.0001

**Table 3 diagnostics-14-00105-t003:** Performance of different deep learning models, evaluated on the UFI test set.

Methods	Models	AUC Score	F1 Score	Kappa Score	Accuracy
Raw data	ResNet152	93.98	84.83	66.62	84.71
	Vision Transformer	85.0	74.59	42.08	76.88
	InceptionResNetV2	84.0	86.85	69.96	86.62
	RegNet	93.96	86.73	70.30	86.84
	ConVNext	**94.44**	**87.04**	**71.10**	**87.05**
Data augmentation	ResNet152	94.9	83.78	63.05	84.71
	Vision Transformer	85.31	77.39	50.03	77.28
	InceptionResNetV2	94.85	76.73	69.25	86.41
	RegNet	95.57	87.20	71.43	87.26
	ConVNext	**96.07**	**88.71**	**77.45**	**89.06**
Proposed system	ResNet152	**96.47**	**89.09**	**75.61**	**89.17**
	Vision Transformer	94.50	87.10	71.35	87.26
	InceptionResNetV2	95.20	88.10	73.50	88.11
	RegNet	96.04	88.40	73.98	88.54
	ConVNext	96.13	88.27	74.98	89.08

## Data Availability

Data available on request (data availability is decided by the IRB of Kangbuk Samsung Hospital on each request).

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
