# Peer review of "Retinal Disease Diagnosis Using Deep Learning on Ultra-Wide-Field Fundus Images"

_diagnostics, 2024, doi:10.3390/diagnostics14010105_

Round 1
Reviewer 1 Report
Comments and Suggestions for Authors
In this study, the authors have compared multiple DL models for diagnosis of abnormalities on retinal images. The article reads well however there are major issues that needs to be addressed for the clarity of the novelty.
Abstract
1. Please clarify which computer vision techniques utilized in the study.
2. Please remove the following statement from abstract, "The system operates through three primary stages: a data augmentation, data preprocessing, and classification."
3. Please mention the tested models, or at least how many models were tested?
4. Please clarify whether AUC value is for training or validation?
5. The abstract should be concise and reflecting the study. In the recent form you use general terms to describe the methods. Please be specific for the methods integrated in the study.
Main text
1. Please extend description of FIgure 1. What the reader expect comparing the two imaging approaches? Higher details, etc.
2. Could you please discuss the cost of the UFI compared to conventional one to further support your claim, "UFIs could play a significant role in telemedicine applications, proving invaluable in regions with limited access to specialized ophthalmological care,"
3. The statement on line 52-53 contradicts with your claim on line 38-39. Please revise it.
4. You may remove the following statement, "The dataset used in our study comprises images procured using these
cutting-edge UFI techniques."
5. Preprocessing of the images using traditional techniques is not a contribution.
6. Data augmentation using traditional transformation is also not a contribution.
7. Have you generated models with/without preprocessing for comparison? Despite the expectation, the supplementary benefit of the preprocessing of the data is not strongly supported.
8. Deep learning models are included millions of neurons that able to learn the characteristics of the data pattern. Therefore, the histogram equalization may not be supporting the model development. Either you should provide stronger proof or you should perform study to demonstrate it.
9. Section 2.3 is described unnecessarily. Please just cite the measurements only.
10. The visualization part is not clearly described.
11. The comparison of the recent study with similar studies is required.
Comments on the Quality of English LanguageIt's fine.
Reviewer 2 Report
Comments and Suggestions for Authors
The study is interesting becouse of its great potential for practical application in the future.
Fundus images offer insights into systemic health conditions, including but not limited to diabetes, oncological disorders, and cerebrovascular events. Ultra-wide-field fundus imaging (UFI) provides comprehensive visualization of eye fundus, including the optic disk, fovea, and macula. evaluated study investigated the application of various deep learning models for detecting eye diseases obtained through UFI, based on the developed an automated system that processes and enhances a dataset of 4,697 images, leveraging advanced computer vision techniques integrated with deep neural networks. The system operates through three primary stages: data augmentation, data preprocessing, and classification. Additionally,the paper presents visualizations of the model’s predictions, including confidence scores and heatmaps that highlight the model’s focal points (particularly where lesions due to damage are evident). By streamlining the diagnosis process and providing intricate prediction details without human intervention, described system serves as a pivotal tool for ophthalmologists. This research underscores the compatibility and potential of utilizing UFI in conjunction with deep learning. The study is interesting becouse of its great potential for practical application in the future.
Author Response
Thank you for your comments.
Reviewer 3 Report
Comments and Suggestions for Authors
-
The paper's literature review lacks depth. The authors should incorporate a minimum of 10 related works, focusing on UFI and closely associated fundus images to enhance the paper's context.
-
Revise the caption for Figure 3 to read: "Figure 3. Different core blocks of the proposed deep learning models."
-
Figure 4's readability is considerably poor due to its small size. Please ensure the figure's legibility is improved for better comprehension.
-
It is recommended to include a succinct section within the Conclusion addressing limitations encountered in the study and suggesting potential directions for future work.
-
For a more comprehensive evaluation, compare your method with at least 4 state-of-the-art methods. Present the comparative results in a tabular format for clarity and better understanding of the model's performance against existing approaches.
Reviewer 4 Report
Comments and Suggestions for Authors
The paper is well drafted and has no such major comments however some points are required to address it properly.
1. Why ResNet152 was selected for study? Why weren't other models chosen?
2. Correlate the research with clinical evidence and pitfalls to overcome in making it patient-oriented.
3. Is there any drug delivery-related information available (For example Occusert and other Controlled release drug delivery system?
Comments on the Quality of English Language
Must be improved.
Round 2
Reviewer 1 Report
Comments and Suggestions for Authors
Thank you for revising the manuscript with the reviewers' comments. I have no further comments.